# Chronic kidney disease causes and outcomes in children: Perspective from a LMIC setting

**Farhana Amanullah[1]⊗, Amyn A. Malik[2,3]⊗\*, Zafar Zaidi[1]**

**1** The Indus Hospital and Health Network, Karachi, Pakistan, **2** Interactive Research and Development (IRD) Global, Singapore, Singapore, **3** Yale Institute for Global Health, New Haven, Connecticut, United States of America

⊗ These authors contributed equally to this work.
\* amyn.malik@yale.edu, amyn.malik@ird.global

**Data Availability Statement:** Patient records cannot be publicly shared due to patient confidentiality as per institutional (Indus Hospital and Health Network) and IRB policy. De-identified data with specified variables will be made available on request from the the institutional review board

## Abstract

### Background and objective

Chronic kidney disease (CKD) constitutes a major public health challenge, with a global prevalence of 15–74.7 cases /million children. Preventing CKD in children, slowing its progression and management of complications are essential, especially in challenged health systems in low middle income countries (LMIC). We conducted a retrospective review to assess the underlying cause and stage of CKD at presentation and clinical outcomes in children and adolescents at the Indus Hospital and Health Network (IHHN) in Karachi, Pakistan.

### Methods

Children 0–16 years with CKD stage 1 and/or higher at presentation were included. Data including demographics, clinical status and lab results at presentation and during follow-up, surgical intervention if any, kidney function at last visit and outcome at last follow-up was recorded.

### Results

A total of 229 children diagnosed with CKD are included in our study. The median age at diagnosis was 10 years with male: female ratio of 1.8:1. Only 5% children presented in stage 1 CKD. The rate of adverse outcomes is 4.5 times higher in children with CKD stage 3–5 compared to early CKD. Congenital anomaly of kidney and urinary tract (CAKUT) was the underlying cause in 49% children. Children with glomerular disease had comparatively worse outcome. Proteinuria, hypertension, anemia and bone disease were associated with high morbidity and mortality.

### Conclusion

The true epidemiology of childhood CKD is unknown in Pakistan. Our cohort showed better CKD outcomes in children diagnosed early with appropriate surgical and medical follow-up.

(irb@ird.global) and the corresponding author (amyn.malik@ird.global).

**Funding:** The author(s) received no specific funding for this work.

**Competing interests:** The authors have declared that no competing interests exist.

Prompt diagnosis, treatment and prevention of progression can be life-saving in our setting. CKD registry data can inform policy changes that can prevent poor outcomes.

## Introduction

Independent of the initial cause, CKD is a clinical syndrome characterized by a gradual irreversible loss of kidney function that can further progress to end stage kidney disease (ESRD) [1]. CKD is a major public health problem worldwide with a global adult prevalence of 8–18% [2, 3] and a pediatric prevalence of 15–74.7 cases per million children [4]. Registry data from European countries shows CKD prevalence of 56–96 per million children [5]. Very little is known about its true burden in children particularly in low resource settings. ESRD is a devastating disorder associated with high morbidity and mortality. Childhood CKD presents clinical features such as growth failure and psychosocial issues that significantly impact quality of life. Cardiovascular complications secondary to CKD lead to morbidity in young adulthood [6]. The real impact of CKD in children in developing countries is unknown as most children in early stages remain undiagnosed and many others die without accessing renal replacement therapy (RRT) which is expensive and difficult to implement in children without the necessary health worker capacity and health systems setup, apart from select tertiary centers.

Children with CKD pose unique challenges to the health system and their providers who must address the primary kidney disorder as well as the many extra-renal manifestations that complicate management [5]. In Pakistan as in other developing countries, a majority of children with CKD present in late stages and ESRD which comprises an age specific mortality rate for children that is 30–150 times higher than for healthy children even in developed countries [7].

According to the International Pediatric Nephrology Association's (IPNA) Renal Replacement Therapy Registry, of the 23000 children on chronic dialysis in 2018, 246 children were registered dialysis patients from Pakistani centers [8]. Centers providing pediatric hemodialysis are limited to four major cities (personal communication), resulting in access barriers for children who live in remote locations and require twice weekly hemodialysis. Pakistan ranked 7[th] highest in terms of number of children on hemodialysis [9]. There is no national pediatric CKD registry in Pakistan and little is known about the epidemiology, risk factors and underlying causes of this potentially devastating condition to inform public health policy makers. Prevention of CKD and its progression in low resource settings will not only save lives but also improve health outcomes and save precious resources.

This study is designed to assess demographics, underlying cause of CKD, clinical stage at presentation, and the association of the underlying cause and stage with clinical outcomes among children registered in the pediatric CKD clinic at The Indus Hospital (TIH) in Karachi, where all services are provided free of cost. We hope to develop a comprehensive program that can be replicated in sites across the Indus Hospital and Health Network (IHHN) to provide patient centered quality care of this complicated condition.

## Methods

Children 0–16 years who presented to TIH with an estimated glomerular filtration rate (eGFR) 60 ml/min/1.73 $^2$ (estimated using modified Schwartz formula 0.413xheight (cm)/ serum creatinine) [10] or lower, CKD stage 1 or higher at presentation (and at a subsequent visit) between 2008 and 2019 were included in this study. Data was collected from the digital

hospital management information system (HMIS) patient records, retrospectively. Baseline demographics, clinical status and laboratory data at presentation, including blood pressure (BP), growth parameters, anemia status baseline and during follow-up and kidney function (serum creatinine) at last visit (2019) were recorded. Final outcomes including stable in follow-up, death, dialysis (RRT) and lost to follow-up were also recorded. For children with a congenital anomaly of the kidney or urinary tract or stone disease, the surgical procedure done was also recorded. All data was fully anonymized at time of data extraction.

## Analysis

Data was collected on Microsoft Excel. We computed frequency of the stage of CKD at presentation and outcomes of patient. Chi square tests were used to analyze the factors associated with poor outcomes. Kaplan Meier (KM) curves and Cox Proportional Hazard Model were used to analyze time to outcome for early stage vs late stage CKD. Data analysis was done using Stata (StataCorp. 2019. Stata Statistical Software: Release 16. College Station, TX: StataCorp LLC.).

## Ethics approval

As the study was a retrospective medical chart review, ethics approval and requirement for informed consent was waived by the institutional review board (IRB) of Interactive Research and Development (IRD).

**Inclusion criteria.** Any child/adolescent 0–16 years of age diagnosed with CKD and followed at the Indus Hospital pediatric nephrology and urology clinic from 2008–2019.

**Exclusion criteria.** Children with Acute kidney injury that reverted to normal kidney function during follow-up.

**Definitions.** CKD stages in children were defined as follows- [11]

*Stage 1*. Kidney damage with a normal or increased GFR (>90 mL/min per 1.73 $m^2$)

*Stage 2*. Mild reduction in the GFR (60 to 89 mL/min per 1.73 $m^2$)

*Stage 3*. Moderate reduction in the GFR (30 to 59 mL/min per 1.73 $m^2$)

*Stage 4*. Severe reduction in the GFR (15 to 29 mL/min per 1.73 $m^2$)

*Stage 5*. Kidney failure (GFR <15 mL/min per 1.73 $m^2$ or dialysis)

*Early CKD*. Stages 1 and 2

*Late CKD/ESRD*. Stages 3, 4 and 5

*CAKUT*. Congenital anomalies of kidney and urinary tract. This category includes obstructive uropathy and reflux nephropathy.

*Glomerular diseases*. Associated with hematuria, proteinuria, hypertension and/or renal dysfunction. This category includes glomerulonephritis and glomerulosclerosis.

*Cystic disease*. Multiple diseases that consist of renal cysts including autosomal recessive and dominant polycystic kidney disease (ARPKD and ADPKD), juvenile nephronophthisis (JNPHP), multicystic dysplastic kidney (MCDK)

*Anemia*. Hemoglobin <11 g/dl and hematocrit <33% [12].

*Mineral bone disorder (MBD)*. A disorder of mineral and bone metabolism due to CKD that is manifested by either one or a combination of the following 1) abnormalities of calcium, phosphorous, PTH or vitamin D metabolism 2) abnormalities in bone histology, linear growth or strength 3) vascular or soft tissue calcification [13].

*Growth failure*. Height less than the third percentile for age or a height standard deviation score (SDS) more negative than -1.88 [14].

## Results

There are a total of 229 children 0–16 years who presented with some degree of persistent kidney dysfunction, were diagnosed with CKD and included in our study. The median age at diagnosis was 10 years (with an IQR of 5–13 years). The median age at outcome was 13 years (with an IQR of 9–15 years). Gender analysis showed that 64.2% children (147/229) were boys with a male to female ratio of 1.8:1.

Non- glomerular disease accounted for the most common underlying condition leading to CKD in our cohort (79%) (Table 1) Whereas glomerular disease was the underlying condition in 15% children and in 5.7% children the underlying renal disease could not be identified. CAKUT was the most common cause of CKD in our cohort (49%) with the most common conditions being obstructive uropathy (23.5%, n = 54) and reflux nephropathy (19%, n = 44) (Table 1)

Most children presented in stages 3–5 of CKD (n = 166; 72.5%) with only 5% (n = 12) presenting in stage 1. Less than a third of the children in our cohort presented in stage 1–2 combined (Table 2).

Analysis of overall outcomes showed that of the cohort only 46.3% children were in stable clinical and or surgical follow-up at the end of the study. Children with poor renal outcomes include 18% on dialysis (although their final outcome at the dialysis facility was not known at the time of study completion). Children lost to follow-up were 25.8% and confirmed dead were 10% (Table 3). We found that 65% (73/112) children presented in earlier CKD, underwent urologic surgery and had a renal survival of 90% (66/73) at 10-year follow-up. Another important finding was that among the children with posterior urethral valves (PUV) (n = 34) that is known to be associated with poorer outcomes, even though 70.5% presented with an eGFR 30 ml/min/1.73m$^2$ and lower, 62% had stable dialysis free outcomes long term.

We compared early CKD stages (1 and 2) with later CKD stages/ ESRD (3, 4 and 5) to evaluate the difference in outcomes if any and created a Kaplan Meier (KM) curve which showed that survival in the first 6 months was similar between the two groups (0.96 vs 0.81) with a sharper drop in the later CKD stages and a survival at 2nd year of 0.91 vs 0.67. The survival rate in the later CKD cohort dropped to nearly half that of the earlier CKD cohort by the 4th year of follow-up (0.90 vs 0.60; Fig 1). The incidence rate of death and loss to follow-up among those with early CKD was 3/100 person years compared to the later stages 16/100 person years.

Children with glomerular disease leading to CKD had the worst outcomes (death or loss to follow-up) compared to children diagnosed with CAKUT who had the most favorable outcomes. Children with stone disease (50% of whom presented in later stages of CKD) had intermediate outcomes (Fig 2).

**Table 1. Underlying diagnosis among children presenting with CKD.**

| Underlying Cause | Diagnosis | N | % |
|---|---|---|---|
| Non glomerular (n: 182/229, 79%) | CAKUT | 112 | 49 |
| | • VUR | 44 | 19 |
| | • Obstructive uropathy | 54 | 23.5 |
| | • UTIs | 12 | 5 |
| | Hypoplastic/dysplastic kidneys | 22 | 9.6 |
| | Stone disease | 40 | 17.5 |
| | Cystic disease | 8 | 3.5 |
| Glomerular disease | Glomerular disease | 34 | 15 |
| | Unknown | 13 | 5.7 |

**Table 2. CKD stage at presentation.**

| CKD stage at presentation | N | % |
|---|---|---|
| 1 | 12 | 5.2 |
| 2 | 51 | 22.3 |
| 3 | 62 | 27.1 |
| 4 | 54 | 23.6 |
| 5 | 50 | 21.8 |

The rate of adverse outcome (death or dialysis) is 4.5 times higher (95% CI: 2.3–8.7) in children with late CKD/ESRD as compared to early CKD after adjusting for age and sex (Table 4). Hazards ratio for age was statistically significant as older age at diagnosis was found to be associated with higher rates of adverse outcomes (Table 4).

Hypertension prevalence was 37% overall increasing to 73% among children in later CKD stages 3–5. Similarly, children with CKD stages 3–5 had an anemia prevalence of 81% (n = 48/64) compared to 33% (n = 52/165) in the earlier CKD group (Table 5). Presence of proteinuria, hypertension, anemia, and mineral bone disorder were statistically significantly associated with adverse outcomes.

## Discussion

The most important finding of our 10-year experience is that delayed presentation at CKD stage 3–5 significantly increases the hazard of poor renal outcomes and mortality. Delayed recognition of CKD mostly results in poor clinical outcomes and significant economic burden on limited health resources.

Almost half the children in our CKD cohort had a congenital anomaly of the kidney and urinary tract as the underlying cause. Although this has been shown from most registry data from developed countries, a possible reason for this finding in our study from a developing country regional center is that the Indus Hospital is a tertiary care referral center with an active pediatric urology service well integrated with nephrology services. CAKUT are common causes of CKD in children, representing 53% of the diagnoses in the Chronic Kidney Disease in Children (CKiD) study as well as the most common cause in the North American Pediatrics Renal Transplant Cooperative study (NAPRTCs) data [15]. The most important conditions in this category in the CKiD data consist of obstructive uropathy (20%) and reflux nephropathy (15%), similar to our cohort. An Indian study from 2017 that reported obstructive uropathy (15%) reflux nephropathy and neurogenic bladder (32%) as the most common causes [16]. Similar findings of predominance of CAKUT are also seen in, the ItalKid Project and the ESCAPE trial [5, 17] as well as a Japanese CKD survey in children where CAKUT represented 68% causes of CKD in children [18].

Our cohort had 15% children with glomerular causes underlying their CKD, very similar to an Indian study that was also set in a tertiary care setting [16]. Interestingly in an Iranian study

**Table 3. Outcome at final follow-up among children who presented with CKD.**

| Outcomes | N | % |
|---|---|---|
| Stable in follow-up | 106 | 46.3 |
| On RRT (dialysis) | 41 | 18 |
| Lost to follow-up | 59 | 25.8 |
| Died | 23 | 10 |

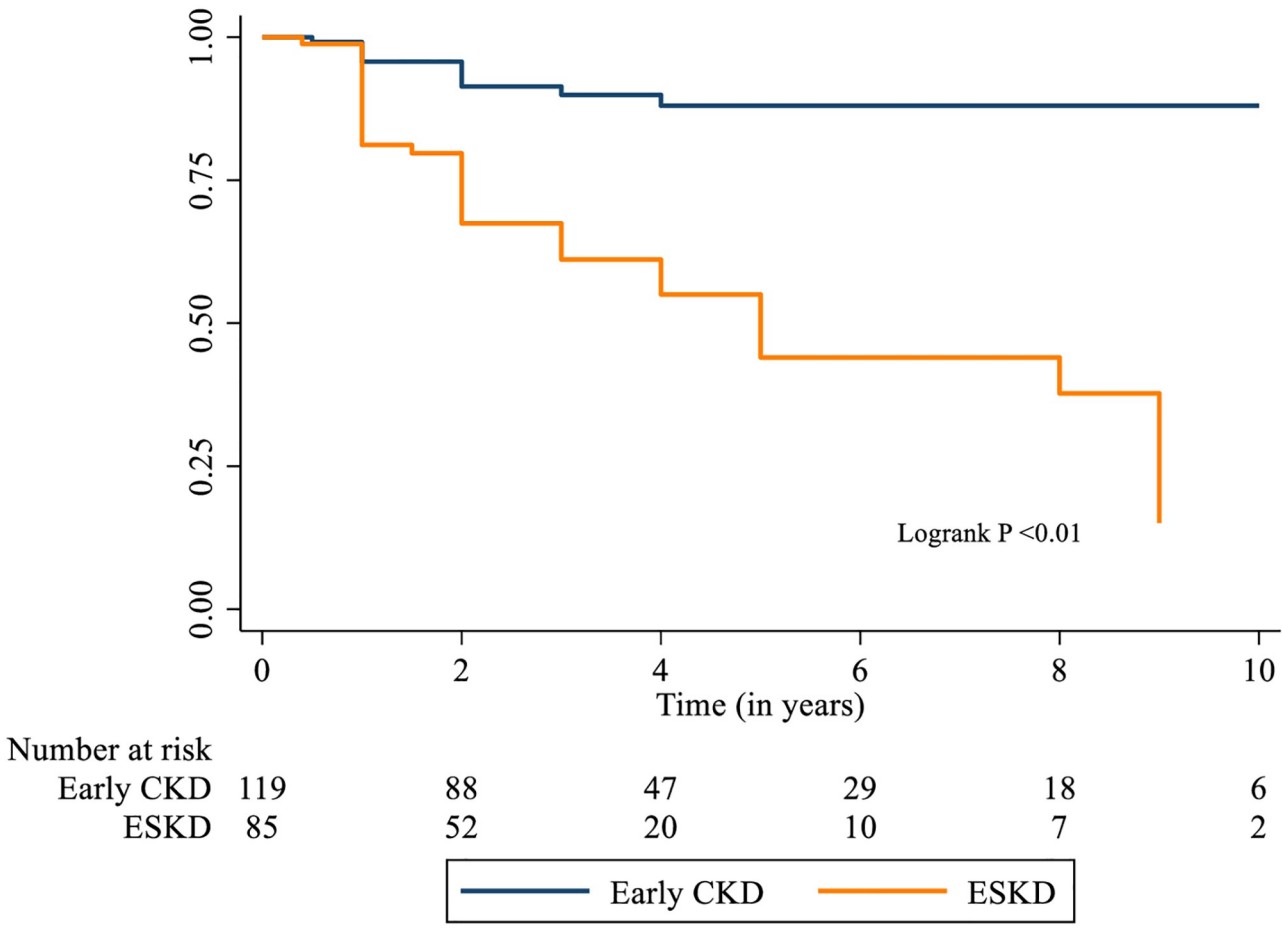

**Fig 1. Survival curve of children with early stages of CKD compared to those with later stages (ESRD).**

with 244 children glomerular disorders accounted for 35.2% of CKD and a study from Nigeria reported 90% prevalence of glomerular diseases among children with CKD. The authors described high infection rates resulting in glomerular diseases and lack of routine prenatal imaging and capacity to detect congenital urologic anomalies as the cause of low prevalence of CAKUT in this group [16]. Additionally urologic anomalies often have subtle initial symptoms that care givers may not notice. This is in contrast to glomerular disease where symptoms (hematuria, edema, oliguria) prompt care givers to seek earlier care. An exclusive renal disease center in Karachi receives the bulk of renal emergencies, which could be a reason for lower glomerular disease prevalence in our cohort. In one study this renal center found that 63.8% (74/116) of renal emergencies in children were due to glomerular disease, also notable was nearly 40% poor renal outcomes (CKD, ESRD and death) in this group [19].

A significant proportion (17.5%) of children in our cohort had urolithiasis. Kidney stones, unfortunately, are a leading cause of renal failure in Pakistan and an increase in renal stone prevalence in children and adults has been noted worldwide [20–23]. The Indus Hospital has a large patient catchment area and provides free of charge services to all patients making it a tertiary care hospital of choice for complicated urolithiasis patients especially children and those requiring multispecialty care. In a study from a major renal center in Pakistan, 67% (n = 242/360) patients with stones presented with renal dysfunction and 32% had poor outcomes

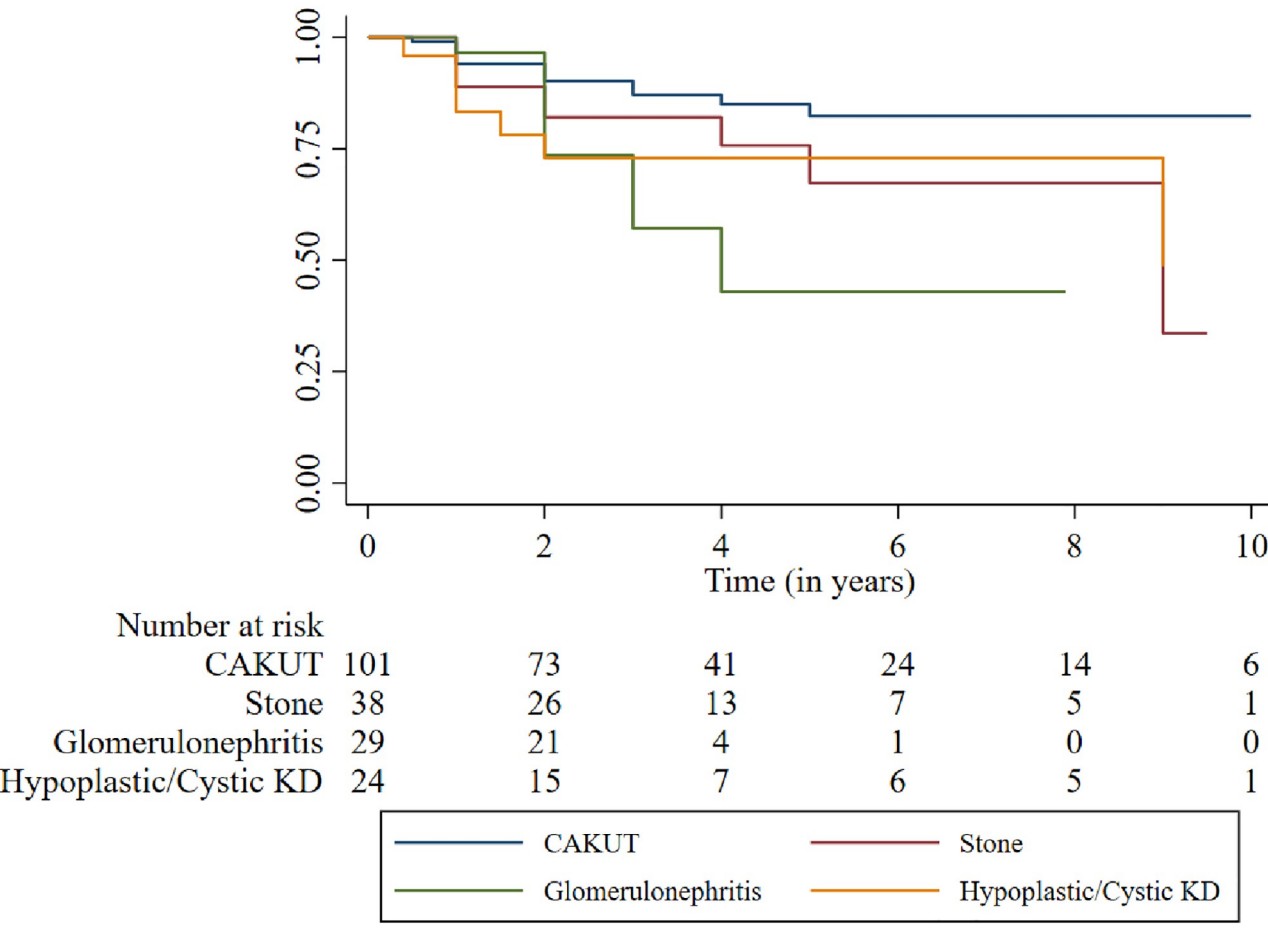

**Fig 2. Survival curves for each underlying disease category in the cohort.**

(dialysis and death). The same center reported that 20% of children who had renal transplants developed ESRD from stone disease [24].

The prevalence of hypertension is tenfold higher with CKD compared to the general pediatric population and is known to increase as children progress through the stages of CKD, so by the time children are on dialysis, 70% will be hypertensive [25]. In our CKD cohort we found a hypertension prevalence of 37% and the prevalence increased to 73% among children in later CKD stages 3–5. The overall prevalence is lower compared to NAPRTCS where 50% children with CKD were hypertensive and CKiD that showed 54% of participants had hypertension at enrolment [26, 27]. As no child had ambulatory BPs performed we do not have estimates of

**Table 4. Unadjusted and adjusted results from Cox Proportional Hazard Model for death or dialysis in patients diagnosed with late CKD/ESRD compared to early CKD.**

|  | Unadjusted | | Adjusted | |
|---|---|---|---|---|
|  | HR | 95%CI | HR | 95%CI |
| Late CKD/ESRD | 5.2 | 2.7–10.0 | 4.5 | 2.3–8.7 |
| Sex (Male) | 0.61 | 0.35–1.1 | 0.78 | 0.44–1.4 |
| Age (1 year) | 1.1 | 1.1–1.2 | 1.1 | 1.0–1.2 |

**Table 5. Prevalence of manifestations of late-stage CKD in our cohort.**

| Risk Factors | CKD 3–5 | | Total |
|---|---|---|---|
| | Yes | No | |
| | n (%) | n (%) | |
| | N = 64 | N = 165 | |
| Proteinuria (N = 210) | 33 (58.9) | 39 (25.3) | 72 |
| HTN (N = 215) | 45 (72.6) | 37 (23.6) | 82 |
| MBD (N = 219) | 38 (67.9) | 55 (34.6) | 93 |
| Anemia (N = 219) | 48 (81.4) | 52 (32.5) | 100 |
| Growth Failure (N = 218) | 50 (86.2) | 120 (55.0) | 170 |

masked or undiagnosed hypertension in this population which may have been high [28]. Presence of hypertension was associated with poorer outcomes in our CKD population.

We found an anemia prevalence of 81% in our late CKD cohort (compared to 33% in the earlier CKD group) and a significant association with dialysis and death. NAPRTCs reports an anemia prevalence of 73% at stage 3 CKD, 87% at stage 4 and >93% at stage 5 [29]. Anemia is known to be a risk factor for poor outcomes including poor growth, neurocognitive problems and is an independent predictor of mortality in adolescents on hemodialysis [30, 31].

An evaluation of overall outcomes showed that only 46% children (82% of whom had non-glomerular CKD) in our cohort were doing well in stable follow-up. More than half the children had poorer renal and possibly survival outcomes, highlighting the terminal nature of childhood ESRD in our setting where services are limited and are not available closer to patient's homes. Even though mortality was 10% we expect the 25% lost to follow-up also had poor outcomes.

CAKUT are the most common cause of ESRD in children but have a very heterogeneous course, with severely affected newborns progressing to ESRD very early on but in most children who are properly managed renal function improves and remains stable at least till puberty [32]. An important finding of our study is that in a pediatric urology referral center such as ours, prompt referral, diagnosis and integrated management (including urologic surgery) may have helped improve outcomes in the non-glomerular/ urologic CKD group. Our finding of 62% of those with PUV having stable dialysis free outcomes long term compares favorably to some studies in literature that found poorer outcomes in PUV-CKD associated with a high nadir creatinine among other variables [33, 34].

In our cohort renal stones are associated with poorer renal survival likely due to late CKD (70%) presentation. Stone disease is a preventable cause of CKD and is not a recognized cause of renal failure in developed countries. Renal stone disease requires urgent public health attention for prevention and early detection in Pakistan, particularly in rural settings.

In our cohort, presence of proteinuria, hypertension, anemia and mineral bone disorder at baseline were statistically significantly associated with poorer renal survival and mortality. It has been shown that especially among glomerular patients, higher levels of proteinuria, hypertension and persistent anemia are significantly associated with CKD progression [35]. Our findings can serve to indicate targets for specific interventions in our population to improve renal outcomes.

Children with growth failure accounted for 78% of the CKD cohort which is much higher than that reported by NAPRTCS (35%). Although there is some correlation between kidney function and poor growth, marked growth retardation is seen at all levels of kidney function in

the NAPRTCS registry. Growth failure requires optimum nutrition and expensive growth hormone treatment which is often absent in our setting [36].

To our knowledge, this is the first study from Pakistan to document the epidemiology of CKD in children and associated risk factors for progression. We were able to follow up 74% of all children registered in the program to provide 10-year outcomes and calculate the associated hazard for CKD presentation at baseline. Our study has some limitations. This is a retrospective study and hence we were not able to explore all associated risk factors due to incomplete or missing data. The study is based in an urban tertiary referral center and hence findings may not be generalizable to the entire population.

## Conclusions

The diagnosis of CKD is linked with a constant risk of progression, a steady decline in kidney function over time to the point of end stage kidneys requiring expensive and challenging renal replacement therapies and high risk of mortality. Further complications of CKD compromise the overall well-being of the child. Early identification of common causes of CKD in children including CAKUT and renal stones is associated with improved renal and overall survival. Our study helps highlight the need for a National childhood CKD registry and public health measures to enhance early diagnosis, management and prevention of progression of CKD In children.

## Supporting information

**S1 Questionnaire. Questionnaire-inclusivity_CKD in children.**
(DOCX)

## Author Contributions

**Conceptualization:** Farhana Amanullah, Amyn A. Malik, Zafar Zaidi.

**Data curation:** Farhana Amanullah.

**Formal analysis:** Farhana Amanullah, Amyn A. Malik.

**Methodology:** Amyn A. Malik.

**Resources:** Farhana Amanullah.

**Writing – original draft:** Farhana Amanullah.

**Writing – review & editing:** Farhana Amanullah, Amyn A. Malik, Zafar Zaidi.

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
