## [Decision Letter · Decision Letter 0]

18 Feb 2022

PONE-D-21-33769Chronic kidney disease causes and outcomes in children: perspective from a LMIC setting.PLOS ONE

Dear Dr. Amyn Abdul Malik

Thank you for submitting your manuscript to PLOS ONE. After careful consideration, we feel that it has merit but does not fully meet PLOS ONE’s publication criteria as it currently stands. Therefore, we invite you to submit a revised version of the manuscript that addresses the points raised during the review process.

We look forward to receiving your revised manuscript.

Kind regards,

Rajendra Bhimma, PhD

Academic Editor

PLOS ONE

Journal Requirements:

2. Please include a complete copy of PLOS’ questionnaire on inclusivity in global research in your revised manuscript. Our policy for research in this area aims to improve transparency in the reporting of research performed outside of researchers’ own country or community. The policy applies to researchers who have travelled to a different country to conduct research, research with Indigenous populations or their lands, and research on cultural artefacts. The questionnaire can also be requested at the journal’s discretion for any other submissions, even if these conditions are not met.  Please find more information on the policy and a link to download a blank copy of the questionnaire here: https://journals.plos.org/plosone/s/best-practices-in-research-reporting. Please upload a completed version of your questionnaire as Supporting Information when you resubmit your manuscrip

Additional Editor Comments (if provided):

Dear Dr. Amyn Abdul Malik

PONE-D-21-33769: Chronic kidney disease causes and outcomes in children: perspective from a LMIC setting.

Thank you for your submission of the above manuscript. It has been reviewed and there are a few concerns raised by the reviewers that need to be addressed. Please respond to these as soon as possible.

Reviewers' comments:

Reviewer's Responses to Questions

**Comments to the Author**

1. Is the manuscript technically sound, and do the data support the conclusions?

Reviewer #1: Yes

Reviewer #2: Yes

2. Has the statistical analysis been performed appropriately and rigorously? 

Reviewer #1: Yes

Reviewer #2: Yes

3. Have the authors made all data underlying the findings in their manuscript fully available?

Reviewer #1: Yes

Reviewer #2: Yes

4. Is the manuscript presented in an intelligible fashion and written in standard English?

Reviewer #1: Yes

Reviewer #2: Yes

5. Review Comments to the Author

Reviewer #1: Thank you for the opportunity to review this interesting paper.

The paper is a retrospective review of 229 children who presented in various stages of CKD.

The introduction adequately sets the background for the study and the method used is appropriate for the report. The findings are in keeping with previously published work and highlight the importance of early presentation to prognosis. The discussion contextualises the results well and the conclusions are inciteful.

I would suggest that the paper be accepted for publication but that a few minor changes are made as follows:

The Introduction

• The statement starting on Line 71 needs referencing

• The statement starting on Line 75 is unclear as to its meaning and needs to be rephrased and referenced

The Discussion

• The sentence beginning on Line 233 does not read well and is too long. It should be rephrased

• The sentence starting on Line 242 does not read well and needs to be rephrased

• The sentence starting on Line 266 has a full stop in error between the word “cohort” and the word “which” (Line 267) and “which” should be spelt with a small case w

The Tables

• I am not sure that the title of Table 5 is correct. I think that the features mentioned would be regarded as manifestations of, rather than risk factors for, late-stage CKD

Reviewer #2: The study is relevant and informative. A letter from the institution allowing the use of anonymized data would strengthen the ethical standards required for the paper. There is a valid rationale for the study. Research questions are plainly identified and acceptable but the research question should mention a comparison between early and late stage chronic kidney disease as well as a comparison of glomerular and non-glomerular disease. These important comparisons are provided in the results. The protocol was technically sound and well planned. The outcome was valid and the stated hypothesis was adequately tested. However, results from the analysis are mentioned under discussion and should be fully reassigned to the results section before they can be discussed later. The discussion should also make methodical reference to the results obtained. The data is available for scrutiny. The methodology is feasible and can likely be reproduced and replicated. The statistical analysis appear valid. Overall an insightful study that adds to the existing body of knowledge.

6. PLOS authors have the option to publish the peer review history of their article (what does this mean?). If published, this will include your full peer review and any attached files.

Reviewer #1: **Yes: **Cecil Levy

Reviewer #2: No

---

## [Author Response · Author response to Decision Letter 0]

15 Mar 2022

March 1, 2022

Dear Dr. Bhimma

We would like to thank you and the reviewers for your careful review of this manuscript and the insightful comments to help improve our work. We have attempted to address all the concerns that have been raised. We believe that the manuscript is stronger as a result. Please see our point-by-point responses below.

As this is review of medical records for which requirement for informed consent was waived, data can be accessed upon email to the corresponding author and the Institutional Review Board (IRB) of IRD at irb@ird.global. 

On behalf of the authorship team,

Amyn Malik

Comments to the Author

Reviewer #1: Thank you for the opportunity to review this interesting paper.

The paper is a retrospective review of 229 children who presented in various stages of CKD.

The introduction adequately sets the background for the study and the method used is appropriate for the report. The findings are in keeping with previously published work and highlight the importance of early presentation to prognosis. The discussion contextualises the results well and the conclusions are inciteful.

Authors: We appreciate your insightful comments on our work and thank you for your time to review it. 

I would suggest that the paper be accepted for publication but that a few minor changes are made as follows:

The Introduction

• The statement starting on Line 71 needs referencing

Authors: Reference to the online registry maintained by International Pediatric Nephrology Association (IPNA) has been added.

• The statement starting on Line 75 is unclear as to its meaning and needs to be rephrased and referenced

Authors: We have rephrased and referenced this statement as follows:

“Pakistan ranked 7th highest in terms of number of children on hemodialysis.(9)”

The Discussion

• The sentence beginning on Line 233 does not read well and is too long. It should be rephrased

Authors: Thank you for the suggestion. We have rephrased this as follows:

“An evaluation of overall outcomes showed that only 46% children (82% of whom had non-glomerular CKD) in our cohort were doing well in stable follow-up. More than half the children had poorer renal and possibly survival outcomes, highlighting the terminal nature of childhood ESRD in our setting where services are limited and are not available closer to patient’s homes.”

• The sentence starting on Line 242 does not read well and needs to be rephrased

Authors: We have rephreased this sentence as follows:

“An important finding of our study is that in a pediatric urology referral center such as ours, prompt referral, diagnosis and integrated management (including urologic surgery) may have helped improve outcomes in the non-glomerular/ urologic CKD group.”

• The sentence starting on Line 266 has a full stop in error between the word “cohort” and the word “which” (Line 267) and “which” should be spelt with a small case w

Authors: Thank you for pointing this out. We have corrected this error. 

The Tables

• I am not sure that the title of Table 5 is correct. I think that the features mentioned would be regarded as manifestations of, rather than risk factors for, late-stage CKD

Authors: Thank you for the suggestion. We agree that these are associations/manifestations of late-stage CKD. We have corrected the title accordingly. 

Reviewer #2: The study is relevant and informative. A letter from the institution allowing the use of anonymized data would strengthen the ethical standards required for the paper. There is a valid rationale for the study. Research questions are plainly identified and acceptable but the research question should mention a comparison between early and late stage chronic kidney disease as well as a comparison of glomerular and non-glomerular disease. These important comparisons are provided in the results. 

Authors: Thank you for these suggestions. We have added the following to the introduction: 

“This study is designed to assess demographics, underlying cause of CKD, clinical stage at presentation, and the association of the underlying cause and stage with clinical outcomes among children registered in the pediatric CKD clinic at The Indus Hospital (TIH) in Karachi, where all services are provided free of cost.”

We have also uploaded a separate IRB waiver letter in the system as well. 

The protocol was technically sound and well planned. The outcome was valid and the stated hypothesis was adequately tested. However, results from the analysis are mentioned under discussion and should be fully reassigned to the results section before they can be discussed later. The discussion should also make methodical reference to the results obtained. 

Authors: Thank you for pointing this out. We have moved all results from the discussion section to the results section and have referenced the discussion accordingly with out results. 

The data is available for scrutiny. The methodology is feasible and can likely be reproduced and replicated. The statistical analysis appear valid. Overall an insightful study that adds to the existing body of knowledge.

Authors: We appreciate your insightful comments on our work and thank you for your time to review it.

---

## [Decision Letter · Decision Letter 1]

21 Apr 2022

PONE-D-21-33769R1Chronic kidney disease causes and outcomes in children: perspective from a LMIC setting.PLOS ONE

Dear Dr. Malik,

Thank you for submitting your manuscript to PLOS ONE. After careful consideration, we feel that it has merit but does not fully meet PLOS ONE’s publication criteria as it currently stands. Therefore, we invite you to submit a revised version of the manuscript that addresses the points raised during the review process.

The reviewers are largely satisfied with the changes made to the manuscript - however, Reviewer 2 has one minor request that should be addressed. Please see the reviewers' comments below.

We look forward to receiving your revised manuscript.

Kind regards,

Natasha McDonald, PhD

Associate Editor

PLOS ONE

Journal Requirements:

Reviewers' comments:

Reviewer's Responses to Questions

**Comments to the Author**

1. If the authors have adequately addressed your comments raised in a previous round of review and you feel that this manuscript is now acceptable for publication, you may indicate that here to bypass the “Comments to the Author” section, enter your conflict of interest statement in the “Confidential to Editor” section, and submit your "Accept" recommendation.

Reviewer #1: All comments have been addressed

Reviewer #2: All comments have been addressed

2. Is the manuscript technically sound, and do the data support the conclusions?

Reviewer #1: Yes

Reviewer #2: Yes

3. Has the statistical analysis been performed appropriately and rigorously? 

Reviewer #1: Yes

Reviewer #2: Yes

4. Have the authors made all data underlying the findings in their manuscript fully available?

Reviewer #1: Yes

Reviewer #2: Yes

5. Is the manuscript presented in an intelligible fashion and written in standard English?

Reviewer #1: No

Reviewer #2: Yes

6. Review Comments to the Author

Reviewer #1: The paper looks great. The only typo left to correct is the reference in the sentence starting on Line 71 'According to the International Pediatric Nephrology Association’s (IPNA)(8) Renal Replacement Therapy Registry......' should be moved to after the word 'centres' on line 73 as follows: "According to the International Pediatric Nephrology Association’s (IPNA) Renal Replacement Therapy Registry, of the 23000 children on chronic dialysis in 2018, 246 children were registered dialysis patients from Pakistani centers.(8)

Reviewer #2: Kindly add the anaemia prevalence results on line 238 and 239 to the results section and add the corresponding discussion around anaemia in the discussion.

The rest of the paper is readily acceptable.

7. PLOS authors have the option to publish the peer review history of their article (what does this mean?). If published, this will include your full peer review and any attached files.

Reviewer #1: **Yes: **Cecil Levy

Reviewer #2: No

---

## [Author Response · Author response to Decision Letter 1]

12 May 2022

April 25, 2022

Dear Dr. McDonald

We would like to thank you and the reviewers for your careful review of the revised manuscript to help improve our work. We have attempted to address all the remaining concerns. We believe that the manuscript is stronger as a result. Please see our point-by-point responses in red below.

On behalf of the authorship team,

Amyn Malik

Comments to the Author

Reviewer #1: The paper looks great. The only typo left to correct is the reference in the sentence starting on Line 71 'According to the International Pediatric Nephrology Association’s (IPNA)(8) Renal Replacement Therapy Registry......' should be moved to after the word 'centres' on line 73 as follows: "According to the International Pediatric Nephrology Association’s (IPNA) Renal Replacement Therapy Registry, of the 23000 children on chronic dialysis in 2018, 246 children were registered dialysis patients from Pakistani centers.(8)

Authors: Thank you. We have corrected the reference placement as advised. 

Reviewer #2: Kindly add the anaemia prevalence results on line 238 and 239 to the results section and add the corresponding discussion around anaemia in the discussion.

The rest of the paper is readily acceptable.

Authors: Thank you. We have added the anemia results and the relevant discussion as follows:

“Similarly, children with CKD stages 3-5 had an anemia prevalence of 81% (n=48/64) compared to 33% (n=52/165) in the earlier CKD group (Table 5). Presence of proteinuria, hypertension, anemia, and mineral bone disorder were statistically significantly associated with adverse outcomes.” (line 181-184)

“We found an anemia prevalence of 81% in our late CKD cohort (compared to 33% in the earlier CKD group) and a significant association with dialysis and death. NAPRTCs reports an anemia prevalence of 73% at stage 3 CKD, 87% at stage 4 and >93% at stage 5.(29) Anemia is known to be a risk factor for poor outcomes including poor growth, neurocognitive problems and is an independent predictor of mortality in adolescents on hemodialysis.(30, 31)” (line 239-244)

---

## [Editor Report · Decision Letter 2]

25 May 2022

Chronic kidney disease causes and outcomes in children: perspective from a LMIC setting.

PONE-D-21-33769R2

Dear Dr. Malik,

We’re pleased to inform you that your manuscript has been judged scientifically suitable for publication and will be formally accepted for publication once it meets all outstanding technical requirements.

Kind regards,

Carla Pegoraro

Division Editor

PLOS ONE

---

## [Editor Report · Acceptance letter]

30 May 2022

PONE-D-21-33769R2 

Chronic kidney disease causes and outcomes in children: perspective from a LMIC setting. 

Dear Dr. Malik:

I'm pleased to inform you that your manuscript has been deemed suitable for publication in PLOS ONE. Congratulations! Your manuscript is now with our production department. 

Kind regards, 

on behalf of

Dr Carla Pegoraro 

Staff Editor

PLOS ONE